# Plant–Microbes Interaction: Exploring the Impact of Cold-Tolerant *Bacillus* Strains RJGP41 and GBAC46 Volatiles on Tomato Growth Promotion through Different Mechanisms

**DOI:** 10.3390/biology12070940

**Published:** 2023-06-30

**Authors:** Abdur Rashid Khan, Qurban Ali, Muhammad Ayaz, Muhammad Saqib Bilal, Taha Majid Mahmood Sheikh, Qin Gu, Huijun Wu, Xuewen Gao

**Affiliations:** 1Key Laboratory of Monitoring and Management of Crop Diseases and Pest Insects, Department of Plant Pathology, College of Plant Protection, Nanjing Agricultural University, Ministry of Education, Nanjing 210095, China; malix.477@hotmail.com (A.R.K.); 2020202068@stu.njau.edu.cn (Q.A.); m.ayazbiotech@gmail.com (M.A.); muhammadsaqib145@njau.edu.cn (M.S.B.); guqin@njau.edu.cn (Q.G.); hjwu@njau.edu.cn (H.W.); 2Key Laboratory of Food Quality and Safety of Jiangsu Province, State Key Laboratory Breeding Base Institute of Plant Protection, Jiangsu Academy of Agricultural Sciences, Nanjing 210014, China; tahamajid1705@yahoo.com

**Keywords:** *Bacillus* strains, tomato plants, volatile organic compounds, GC-MS analysis, PGPR, antioxidant enzymes

## Abstract

**Simple Summary:**

Microbial volatile organic compounds (VOCs) play a crucial role in promoting plant growth and causing systemic resistance to a variety of diseases caused by fungus, bacteria, nematodes, and oomycetes. However, the role of *Bacillus* VOCs in growth promotion is still limited. In the present work, we aim to examine the growth promotion mechanisms of cold-tolerant *Bacillus* strains RJGP41 and GBAC46 from the Qinghai-Tibet Plateau and the well-known PGPR strain FZB42 and their VOCs on tomato plants. Our experiment results revealed that both *Bacillus* isolates and their pure VOCs positively improve PGPR activities in tomato plants by triggering antioxidant enzyme activity and expression of the PGPR genes. In our future research, the selected *Bacillus* strains and their novel pure VOCs will be further explored to find the possible mechanisms for the safe and green control of tomato disease in sustainable agriculture.

**Abstract:**

The interaction between plant and bacterial VOCs has been extensively studied, but the role of VOCs in growth promotion still needs to be explored. In the current study, we aim to explore the growth promotion mechanisms of cold-tolerant *Bacillus* strains GBAC46 and RJGP41 and the well-known PGPR strain FZB42 and their VOCs on tomato plants. The result showed that the activity of phytohormone (IAA) production was greatly improved in GBAC46 and RJGP41 as compared to FZB42 strains. The in vitro and in-pot experiment results showed that the *Bacillus* VOCs improved plant growth traits in terms of physiological parameters as compared to the CK. The VOCs identified through gas chromatography-mass spectrometry (GC-MS) analysis, namely 2 pentanone, 3-ethyl (2P3E) from GBAC46, 1,3-cyclobutanediol,2,2,4,4-tetramethyl (CBDO) from RJGP41, and benzaldehyde (BDH) from FZB42, were used for plant growth promotion. The results of the partition plate (I-plate) and in-pot experiments showed that all the selected VOCs (2P3E, CBDO, and BDH) promoted plant growth parameters as compared to CK. Furthermore, the root morphological factors also revealed that the selected VOCs improved the root physiological traits in tomato plants. The plant defense enzymes (POD, APX, SOD, and CAT) and total protein contents were studied, and the results showed that the antioxidant enzymes and protein contents significantly increased as compared to CK. Similarly, plant growth promotion expression genes (*IAA4, ARF10A, GA2OX2, CKX2,* and *EXP1*) were significantly upregulated and the *ERF* gene was downregulated as compared to CK. The overall findings suggest that both *Bacillus* isolates and their pure VOCs positively improved plant growth promotion activities by triggering the antioxidant enzyme activity, protein contents, and relative gene expressions in tomato plants.

## 1. Introduction

The rhizosphere is closely packed with microorganisms, having high competition for space and nutrients [1]. Many factors, such as plant roots and soil management, influence these soil microbiotas. To help plants absorb nutrients, plant roots exude primary and secondary metabolic compounds and macromolecules along with cells into the rhizosphere to modify the local microbiota [2]. Root exudates contain organic substances that attract microbes and move toward roots via chemotaxis, thus aiding the plant [3]. These exudates may also protect plants from infections by altering the composition of the local soil microbiota [4]. Furthermore, certain microorganisms secrete compounds that are beneficial to them [5]. Plant growth-promoting rhizobacteria (PGPR) are a diverse group of bacteria that promote plant growth by colonizing the plant roots with a variety of mechanisms, such as phytohormone synthesis and nutrient uptake [6,7]. Many PGPR can also inhibit phytopathogens by releasing antibiotics [8,9,10] or by activating (priming) plant innate immunity, through a process generally called induction of systemic resistance (ISR) [11].

*Bacillus* isolates can promote plant growth through direct and indirect mechanisms [12]. The direct mechanism involves the induction of phytohormones (indole-3-acetic acid (IAA), cytokinin (CT), gibberellic acid (GA), and siderophores), and moreover, solubilizes phosphorus and other nutrients involved in plant growth and root mass [13,14]. Indirectly, the *Bacillus* spp. minimizes the threats that pathogens create during plant growth and development, thereby inducing plant growth [15]. Plant root architectural changes also improve the plant’s capability to utilize soil by enhancing water and nutrient uptake [16].

Several PGPR release volatile organic compounds (VOCs) without coming into physical contact with the plant roots, and stimulate plant growth [2,17,18,19], whereas some PGPR inhibit fungal growth by releasing VOCs [20,21]. Bacterial VOCs are well-known to play various roles, including signaling for inter- and intra-species communication, cell-to-cell communication, stimulating or inhibiting plant growth, and influencing phytopathogens [22]. Some PGPR, including *B. amyloliquefaciens* 1N937a, *Paenibacillus polymyxa* E681, *Bacillus Subtilis* GB03, and *Bacillus atrophaeus* GBSC56, have been found to produce VOCs that stimulate *Arabidopsis thaliana* and tomato growth and development [18,23,24]. More than 25 VOCs are emitted from *B. subtilis* GBO3, which activate transcripts in *A. thaliana* primarily involved in metabolism, hormone regulation, cell wall modification, and protein synthesis [25]. Several microorganisms have previously been shown to produce VOCs that directly antagonize *Rhizoctonia solani* and protozoan development [26]. Bacterial VOCs can have a variety of chemical structures, such as benzaldehyde, amines, benzothiazole, decanal, dimethyl trisulfide, cyclohexanol, 2-ethyl-1-hexanol, and nonanal, being found as fungicidal agents [27].

Previous research demonstrated that *Bacillus* strains can couple with various plants to provide abiotic stress tolerance or disease suppression [28,29] based on the extent of the strain, but it was frequently observed that all the *Bacillus* strains promote plant growth. PGPR in direct interaction have been thoroughly investigated for their ability to improve plant growth and trigger resistance against various plant pathogens [30]. On the contrary, microorganisms capable of producing VOCs in soil or other growth media have gained greater attention for their growth-promoting activities. Our lab has thoroughly investigated *Bacillus* strains collected from Qinghai, Tibet (China) [31], for their antagonistic potential against different pathogenic bacteria and fungi; however, very little has been investigated about the function of the VOCs emitted from these strains on plant growth promotion (PGP) activities in tomato plants. The current study aims to identify effective VOCs produced by *Bacillus* spp. RJGP41 and GBAC46 and their PGP activity in greenhouse and in vivo experiments. Furthermore, the study also adds new insights about the mechanism of *Bacillus* VOCs through the regulation of critical genes involved in promoting tomato plant growth.

## 2. Materials and Methods

### 2.1. Bacillus Strains, Plant Material, and Culture Conditions

The *Bacillus* sp. RJGP41 and *Bacillus thuringiensis* GBAC46, previously isolated from our lab (Laboratory of Biocontrol and Bacterial Molecular Biology) [31], and the positive control PGPR strain *Bacillus velezensis* FZB42 [32], were used in this study. The bacterial strains were cultured overnight at 37 °C in Luria-Bertani (LB) medium, and the stock cultures were kept at −80 °C in LB broth containing 30% glycerol. Tomato seeds were firstly washed with 70% ethanol and soaked in a 50% solution of sodium hypochlorite (NaOCl) for 15 min, followed by 4–5 rinses with distilled water (ddH_2_O). The disinfected seeds were placed in petri plates containing 0.5× Murashige and Skoog (MS) medium (pH 5.7) [33] supplemented with 1.5 g of sucrose and agar (0.08%) for the in vitro growth assay.

### 2.2. Production of Indole 3-Acetic Acid (IAA) 

Indole 3-acetic acid (IAA) production was assessed using a calorimetric assay in culture broth [34]. Briefly, the *Bacillus* strains were grown in 100 mL Erlenmeyer flasks containing LB broth and supplemented with 0.1 g/L of l-tryptophan. The culture was incubated at 37 °C for 7 days in a rotary shaker at 200 rpm, both with and without tryptophan. The *Bacillus* strain cultures were centrifuged at 10,000 rpm for 10 min at 4 °C, and the culture filtrate was then collected and passed through a 0.45 μm cellulose filter. Then, 2 mL of the culture filtrate was mixed with 2 mL of the Salkowski reagent (12 g/L of FeCl_3_ and 8 M H_2_SO_4_) and stored in the dark for 30 min. Post-incubation, the presence of pink color was assessed as IAA production, and OD_600_ = 535 nm was examined in an ELISA spectrophotometer (BioRad, Hercules, CA, USA). 

### 2.3. In Vitro Growth Promotion Assay by Bacillus spp.

The effect of *Bacillus* RJGP41, GBAC46, and PGPR strain FZB42 VOCs on tomato growth promotion was examined in partitioned (I-plate) petri dishes. One side of the compartment plate was filled with 0.5× Murashige and Skoog media (MS), pH 5.7 [33], supplemented with 1.5% sucrose and 0.8% agar, whereas the other compartment was filled with LB agar media. Four-day-old, primed tomato seeds (5 in each petri plate) were placed on 0.5× MS media, whereas in other compartments, 5 µL of selected overnight cultured *Bacillus* strains (10^7^ CFU/mL) were inoculated in triplicate. For control plates, 5 µL of LB was used instead of *Bacillus* culture. The petri plates were firmly wrapped with parafilm and incubated in a growth chamber at 25 °C for 14 days under a 16 h light/8 h dark photoperiod. After incubation, tomato seedlings were evaluated for growth promotion traits.

### 2.4. In Planta Growth Promotion Assay by Bacillus spp.

To validate in vitro test results that VOCs are the main regulators in the enhancement of PGP, an in-pot experiment system developed in [35] was used. Tissue culture glass jars (12 cm × 10 cm) were sterilized, and a small petri plate inoculated with 200 µL of the selected *Bacillus* strains (RJGP41, GBAC46, and the positive control FZB42) was placed in the center of the glass jar. Six small holes (2 mm) were made in the bottom of the plastic pots, and a filter paper was inserted in the bottom of each pot. These pots were filled with sterilized peat moss, and three equal-sized tomato seedlings were sown in each plastic pot. The pots were then fixed on tissue culture glass jars and tightly wrapped with parafilm so the *Bacillus* VOCs could not escape. The plastic pots were then stored at 25 °C in a growth chamber for 6 weeks under a 16 h light/8 h dark photoperiod. After 6 weeks, physiological parameters of PGP, such as shoot length (cm), root length (cm), fresh shoot weight (g), fresh root weight (g), shoot dry weight (g), and root dry weight (g), exposed to *Bacillus* VOCs and the control were measured. In each of the experiments, FZB42 served as a positive control. The experiment was replicated thrice.

### 2.5. GC-MS Analysis of Bacillus spp. VOCs 

*Bacillus* strains RJGP41 and GBAC46 were grown at 37 °C and 200 rpm overnight, and 20 µL of the *Bacillus* spp. culture was inoculated in 30 mL of MS agar medium in 100 mL VOC-collecting vials. The vial caps were firmly sealed with 5 parafilm layers and incubated for 5 days at 37 °C. As a control, MS agar medium was used. To collect VOCs, a 2 cm solid-phase micro-extraction (SPME) fiber made of divinyl benzene/carboxen/PDMS (DCP, 50/30μm) (Supelco, Bellefonte, PA, USA) was employed. The SPME fiber was injected into vials containing *Bacillus* strain cultures and incubated at 50 °C for half an hour. Gas chromatography-mass spectrometry (GC-MS) analysis was carried out utilizing a Bruker 450-GC gas chromatograph in conjugation with a Bruker 320 mass spectrophotometer (MS) [36]. The SPME fiber was run via the following program: desorption for 5 min at 220 °C, with a starting column temperature of 35 °C for 3 min, increased to 180 °C at a rate of 10 °C/min, then further increased to 240 °C at 4 °C/min, and held for 5 min. The GC-MS was run for 37 min at a flow rate of 1 mL min^−1^ with helium gas as a carrier. The MS was operated at 70 eV in electron ionization mode with continuous scanning from 50 *m*/*z* to 500 *m*/*z* at a temperature source of 220 °C. The NIST/EPA/NIH Mass Spectral Libraries were used to investigate the mass spectra of the volatile compounds. The VOCs detected as being emitted from RJGP41 and GBAC46 were purchased from Sigma-Aldrich. The strain FZB42 pure VOCs previously identified in [37] and purchased from the same company were used in this study to test for their PGP activity.

### 2.6. In Vitro Plant Growth Promotion by Pure VOCs 

The pure VOCs detected by GC-MS analysis were tested for their ability to promote tomato plant growth traits. VOCs having the highest probability, namely 2,2,4,4-tetramethyl-1,3-cyclo butanediol (CBDO) from RJGP41, 3-methyl-2-pentanone (2P3E) from GBAC46, and benzaldehyde (BDH) from FZB42, were used to evaluate their impact on tomato plant growth. Tomato seeds were surface-sterilized as previously described and grown in 0.5× MS salt media for 3 days in a half-section of an I-plate [33]. The pure VOCs were diluted in methanol [38,39] to a final concentration (of 0, 50, 100, 200, 300, and 400 μg/mL) and applied to another half-section of an I-plate. The petri plates were tightly wrapped with parafilm and incubated at 25 °C for 2 weeks under a 16 h light/8 h dark photoperiod. Sterilized distilled water served as a control (CK). Each treatment was repeated three times, and the experiment was replicated thrice.

### 2.7. In Vivo Plant Growth Promotion by Bacillus VOCs

Plant growth promotion by VOCs (2P3E, CBDO, and BDH) was evaluated as described earlier in PGP by pure VOCs in a greenhouse experiment. After 6 weeks of incubation, PGP parameters, such as shoot length (cm), fresh shoot weight (g), and root morphological studies, were assessed, by using a Rhizo-scanner (EPSON Perfection V700 Photo, Los Alamitos, CA, USA) equipped with WinRHIZO software accessible by Regent Instruments Co. (Québec, QC, Canada), between plants exposed to pure VOCs and the control. Each treatment as well as the experiment were replicated thrice.

### 2.8. Determination of Antioxidant Enzyme Activity and Total Protein Contents

The activity of antioxidant enzymes, such as peroxidase (POD), superoxide dismutase (SOD), catalase (CAT), and ascorbate peroxidase (APX), was evaluated after exposure to pure VOCs, (2P3E, CBDO, and BDH) in tomato plants after 3 weeks of inoculation (dpi) following the method in [18]. Fresh leaf samples (0.3 g) were ground in phosphate buffer solution (PBS) containing 1 mM of EDTA and pH 7.8 in an ice bath. The samples were centrifuged at 12,000 rpm for 30 min at 4 °C and the supernatant served as an enzyme extract and was measured by spectrophotometers. The ddH_2_O-treated leaf extracts were used as a control. POD activity was calculated using Ali et al.’s approach: H_2_O_2_, PBS, enzyme extract, and 3.0 mL of reaction mixture were used in the experiment. At 470 nm, a spectrophotometer was used to measure the mixture’s absorbance [40]. According to the procedure of Wu et al., CAT enzyme activity was measured. H_2_O_2_, PBS, and enzyme extract were included in the 3.0 mL reaction solution, and the absorbance was measured using a spectrophotometer for 3 min at 240 nm [41]. According to Ayaz et al., SOD was tested. H_2_O_2_, PBS, and enzyme extract were included in the 3.0 mL reaction mixture, and the absorbance was assessed using a spectrophotometer at 560 nm [18]. According to Ali et al., APX was investigated. H_2_O_2_, PBS, and enzyme extract were included in the reaction solutions, and absorbance was assessed using a spectrophotometer at 290 nm [42]. In addition to the leaf samples already ground in phosphate buffer solution (PBS) containing 1 mM of EDTA and pH 7.8, protein contents were also examined. The determination of the total protein content was performed as described by Bradford using a calibration curve of bovine serum albumin (BSA) [43].

### 2.9. Relative Expression of Growth Promotion Genes Exposed to Pure VOCs 

Plant samples were collected after 5 days of pure VOCs’ exposure to tomato plants. RNA was extracted using TRIzol reagent (Invitrogen Biotech Co., Waltham, MA, USA) according to the manufacturer’s guidelines and cDNA was synthesized using reverse transcriptase (RT) (Takara Bio Inc., Tokyo, Japan). The sequence of PGP genes (*IAA4, ARF10A, GA2OX2, ERF, CKX2,* and *EXP1*) were obtained from NCBI GenBank and the primers were designed using the Primer Quest tool of Integrated DNA Technologies (Appendix A). The qRT-PCR analysis was performed using a QuantStudio RT Thermocycler (Thermo Fisher Scientific, Waltham, MA, USA) with the chamQ SYBR green qRT-PCR master mix (Vazyme Co., Ltd., Nanjing, China). The conditions employed for qPCR were: initial denaturation at 95 °C for 30 s, 40 cycles for 10 s at 95 °C, and 30 s at 60 °C. The expression of genes under investigation was determined using the threshold (Ct) value for each gene and the Ct value of actin in tomato as a constitutive reference gene [44]. 

### 2.10. Statistial Analysis 

All experiments (in vitro and planta) were performed in a completely randomized design (CRD) and repeated three times. All the data were statistically analyzed using the statistical software SPSS-21.0. All means were separated by using Tukey’s HSD test at *p* ≤ 0.05 after the ANOVA. Origin graphics and analysis software (Version 2023, OriginLab Corporation, Northampton, MA, USA) was used for graphical representations.

## 3. Results

### 3.1. Production of Indole 3-Acidic Acid (IAA)

The progression of pink color in the culture broth due to the presence or absence of tryptophan indicated the production of indole 3-acidic acid (IAA) (Figure 1A). Both *Bacillus* isolates (GBAC46 and RJGP41) and the positive control (FZB42) showed high pink color formation as compared to the negative control (Figure 1A). IAA production in RJGP41 and the positive control FZB42 was similar at OD_600_ = 535 nm (0.16), whereas the GBAC46 was lower at OD_600_ = 535 nm (0.15 as compared to the control (Figure 1B)). The overall production of pink color in the culture broth revealed that both *Bacillus* strains have the ability to produce IAA, having a crucial role in plant growth promotion traits. 

### 3.2. In Vitro Plant Growth Promotion by Bacillus VOCs

Four-day-old, primed tomato seeds (5 in each petri plate) were placed on 0.5× MS media, whereas in other compartments, 5 µL of selected overnight cultured *Bacillus* strains (10^7^ CFU/mL) were inoculated. The results of the selected *Bacillus* strain VOCs showed that a significant growth improvement was observed in terms of PGP traits (fresh and dry seedlings’ weight) in I-compartment plates at 25 °C for 12 days (Figure 2A). In addition, inoculation of the RJGP41, GBAC46, and FZB42 increased the seedling fresh weight by 88.7%, 89.0%, and 87.5% and the seedling dry weight by 19.6%, 19.5%, and 16.8%, respectively, as compared to the CK (Figure 2B).

### 3.3. In Vivo Plant Growth Promotion by Bacillus VOCs

In plastic pot experiments, the application of the selected *Bacillus* VOCs produced by the selected *Bacillus* strains significantly increased tomato plant growth as compared to the control (CK). The selected *Bacillus* VOCs emitted by GBAC46, RJGP41, and FZB42 improved plant growth attributes such as shoot length (45.29%, 40.93%, and 41.21%) and root length (31.63%, 32.35%, and 28.68%), respectively, as compared to the CK. Whereas, the fresh shoot weights by GBAC46 (45.5%), RJGP41 (44.6%), and FZB42 (41.2%), and the fresh root weights by GBAC46 (38.5%), RJGP41 (39.5%), and FZB42 (37.3%) were observed compared to the CK (Figure 3).

### 3.4. GC-MS Analysis of Pure VOCs Emitted by Bacillus Strains

PGP characteristics revealed that VOCs emitted by RJGP41, GBAC46, and FZB42 play an important role in plant growth stimulation. The VOCs emitted by RJGP41 and GBAC46 were detected through GC-MS analysis and selected in a time duration from 0 to 30 min based on the previous results. The mass spectra data of the possible VOCs emitted by *Bacillus* strains were compared to the data from the NIST/EPA/NIH Mass Spectral Libraries. Here, 12 VOCs from GBAC46 and 11 VOCs from RJGP41 were detected through GC-MS analysis. The VOCs emitted from GBAC46 had a retention time between 3 and 20 min, with a MW ranging from 68 to 645. Although, VOCs emitted by RJGP41 had a retention time ranging from 3 to 19 min and a MW ranging from 112 to 5781. Both GBAC46 and RJGP41 had area ranges of 1.268 × 10^9^ to 7.125× 10^9^ and 1.368 × 10^9^ to 9.852 × 10^9^, respectively (Figure 4). The VOCs similar to the control were ignored and not selected. Each VOC had a probability percentage, CAS number, molecular weight (MW), and chemical formula identified (Table 1).

### 3.5. In Vitro Plant Growth Promotion by Pure VOCs

The pure VOCs detected through GC-MS analysis were evaluated for tomato PGP in vitro. There was a high probability of VOCs, namely 2,2,4,4-tetramethyl-1,3-cyclobutanediol (CBDO) from RJGP41 and 3-methyl-2-pentanone (2P3E) from GBAC46. For the positive control strain FZB42, the VOC benzaldehyde (BDH), already reported by [19], was used in this study. These pure VOCs were purchased from Sigma Aldrich and screened independently for their PGP traits, with different concentrations (0, 50, 100, 200, 300, and 400μg/mL). In I-plate experiments, the results indicated that the maximum increase in growth promotion of tomato seedlings was observed in 2P3E-VOC from GBAC46, with a fresh seedlings’ weight of 44.01% and dry seedlings’ weight of 43.97%, respectively, as compared to CBDO-VOC (37.86% and 35.32%) and BDH-VOC (34.59% and 43.33%) at 200 μg/mL (Figure 5). The CBDO-VOCs from RJGP41 showed higher growth promotion in the fresh seedlings’ weight (44.47%) and dry seedlings’ weight (46.57%), as compared to BDH-VOCs (36.97% and 46%) and 2P3E (15.21% and 10.66%) at 300 μg/mL. The higher concentrations (300 and 400 μg/mL) of 2P3E and CBDO-VOCs reduced the seedlings’ growth promotion traits (Figure 5).

### 3.6. In Vivo Plant Growth Promotion by Pure VOCs

The pure VOCs BDH, 2P3E, and CBDO significantly increased the PGP attributes, such as shoot length (cm) and shoot weight (g), in the pot experiment as compared to the CK plants. The maximum shoot length (52.5%) was recorded by 2P3E-VOC, followed by CBDO-VOC (51.3%) and BDH-VOC (48.7%), respectively. Similarly, the highest shoot weight was achieved by CBDO treatment (29.1%), followed by 2P3E (26.87%) and BDH (13.25%), respectively, as compared to the CK. Our results indicated that the selected VOCs play a crucial role in PGP in tomato seedlings as compared to the CK (Figure 6).

Post-exposure to pure VOCs, rhizo-scanning studies revealed a considerable increase in the root morphological parameters. All the root morphological parameters, such as root length, root tips, root average diameter, root average volume, and root surface area, were found to be significantly maximum in pure VOC exposure treatments as compared to non-exposed treatments (Figure 7). Root length was the maximum (35.46%) in the 2P3E-VOC treatment, while all other parameters: root average volume (63.21%), root average diameter (37.1%), root tips (56.60%), and surface area (62.56%), were maximum in the CBDO-VOC treatment, followed by the BDH treatment (32.41%, 58.1%, 35.72%, 49.09%, and 59.23%). Overall, the data revealed that the CBDO treatment highly promoted PGP activities, followed by 2P3E and BDH, respectively. In comparison to the CK, the selected pure VOCs showed improved root morphological traits, indicating a key role for pure VOCs in PGP traits. 

### 3.7. Antioxidant Activity and Total Protein Contents Post-Exposure to Pure VOCs in Tomato Plants

The activity of plant defense enzymes (POD, CAT, SOD, and APX) was evaluated after exposure to the pure VOCs 2P3E, CBDO, and BDH in tomato plants. The results indicated that the activity of antioxidant enzymes was increased in the selected pure VOCs treatments as compared to the CK. The SOD significantly increased in CBDO, 2P3E, and BDH by 67.75%, 46.04%, and 38.38% and in POD by 59.24%, 55.43%, and 48.10%, respectively, as compared to the CK. The treatment of the pure VOCs CAT and APX was also examined, and the results showed that the activity of CAT was increased by CBDO (36%), 2P3E (34.53%), and BDH (31.81%), and APX by CBDO (62.03%), 2P3E (43.12%), and BDH (37.79%), respectively, as compared to the CK. In addition, the total protein content was observed, and the results showed that the application of pure VOCs significantly enhanced the protein content in tomato plants compared to the CK (Figure 8E). The overall experiment data revealed a considerable stimulation in the activity of all enzymes and total proteins post-exposure to pure VOCs (Figure 8).

### 3.8. Effect of Pure VOCs on Plant Growth Promotion Genes

The expression of growth-promoting genes in tomato plants was studied after exposed to the pure VOCs 2P3E, CBDO, and BDH. The results illustrated that the selected pure VOCs significantly upregulated the growth promotion genes (*IAA4, ARF10A, CKX2, GA2OX2*, and *EXPI*), whereas the *ERF* gene was downregulated (Figure 9). The relative expression of the *ERF* gene was downregulated after exposure to 2P3E. The most significant results were obtained with the 2P3E VOC in growth promotion genes (*IAA4, ARF10A, CKX2,* and *GA2OX2*), which were significantly upregulated, followed by CBDO; on the other hand, the expression of the *EXP1* gene was highly regulated in CBDO, followed by BDH. Overall, the experiment results revealed that plant growth-promoting genes in tomato leaves were highly regulated in the selected treatments as compared to the CK.

## 4. Discussion

Microbial VOCs produced by *Bacillus* species have been proven to play an important role in triggering induced systemic resistance (ISR) against many plant pathogens and enhancing plant growth [17,37,40,45]. VOC application is considered a safe and environment-friendly method, as this method utilizes physically contact-free interactions with the pathogen or plant [46,47]. Previously, most studies described the in vitro interaction of VOCs with plants [24,48]. Park et al. reported that bacterial VOCs are capable of stimulating PGP in vitro as well as in vivo [35]. Thus, in the present study, we demonstrated the influence of *Bacillus*-produced VOCs on PGP in tomato plants. Previous studies demonstrate that PGP in different plants occurs through direct plant hormone production, and subsequently, nutrient uptake [49,50,51]. Our results clearly demonstrated PGP in tomato plants, mediated by *Bacillus* VOCs. Both *Bacillus* isolates, GBAC46 and RJGP41, were screened for PGP activity and showed IAA production, indicating growth promotion characteristics. Among the benefits of IAA production in culture by *Bacillus* isolates, the foremost direct benefit is root and shoot elongation, particularly when the *Bacillus* interacts with hormone transportation within plants [25]. Although IAA is more involved in plant growth mechanisms, it may play a crucial role in plants stress tolerance [52].

In nature, plants can sense environmental stresses and microbial VOCs and respond to them. This helps them control their growth and build up systemic resistance [46]. In our study, in vitro and in vivo results indicated that when tomato plants were exposed to *Bacillus*-produced pure VOCs, a substantial increase in PGP was observed. *Bacillus*-produced pure VOCs significantly increased the fresh plant weight and dry plant weight in I-compartment petri plates (in vitro), and the root length, shoot length, root weight, and shoot weight in plastic culture tubes (in planta). The finding of Qurban et al. also demonstrates the role of *Bacillus*-produced pure VOCs in PGP, which is in accordance with our findings of PGP by *Bacillus*-produced pure VOCs [2]. Fincheseira et al. reported similar findings, stating that the BCT9 strain produces VOCs such as 3-hydroxy-2-butanone, 2,3-butanediol, 2-nonanone, 2-tridecanone, and 2-pentadecanone, which play an important role in PGP [45].

To investigate the effectiveness of pure VOCs on tomato PGP, the I-plate system was utilized. The pure VOCs in each *Bacillus* isolate were tested at different concentrations (0, 50, 100, 200, 300, and 400 μg/mL). Our experiment findings indicated that the highest PGP was observed when 2P3E, CBDO, and BDH were used as compared to the CK. The higher concentrations (300 μg/mL and 400 μg/mL) of VOCs had a negative impact on the PGP activity of tomato plants. According to the current results, optimum concentrations of VOCs promote plant growth, but higher concentrations are toxic to plants. Different studies indicate that *Bacillus* VOCs, when applied at a concentration of ≤400 μg/mL, significantly promote plant growth in different crops [2,18,19,46]. Higher concentrations may be toxic for the plant but the best for fungi due to the chemical nature of the VOCs [23]. These findings suggested that the VOCs (2P3E, CBDO, and BDH) are primarily involved in the PGP of tomato plants.

Growth conditions in petri plates and in the field greatly vary in terms of growth media and the extent of the interactions with different environmental factors. Therefore, we investigated the effect of pure *Bacillus* VOCs in planta to check PGP attributes in tomatoes. In planta, PGP by pure *Bacillus* VOCs is documented in many previous studies [18,19,53,54,55]. Volatile emissions from *B. amyloliquefaciens* GB03 have been reported to accelerate the ability by increasing the chlorophyll contents and photosynthesis efficiency [56]. Park et al. also revealed that *P. fluorescens* SS101-produced VOCs might increase plant growth and development [35]. Our findings revealed that pure *Bacillus* VOCs might increase numerous plant growth parameters in tomato plants by producing diverse VOCs via a non-contact co-culture system of *Bacillus* and plants. Our results revealed a clear and significant increase in the plant weight, fresh shoot weight, dry shoot weight, shoot length, fresh root weight, and dry root weight of tomato plants after exposure to *Bacillus* pure VOCs.

It is well-established that the root morphological characteristics are the basic components of the plant root system and play a unique role in nutrient absorption in plants [57]. Gutierrez-Luna et al. carried out an experiment and investigated that *Bacillus* species are mainly involved in the modification of the root architecture, elicitation of primary root length, lateral root number, total fresh weight, and length in *Arabidopsis thaliana* [58]. We also investigated whether the VOCs produced by GBAC46, RJGP41, and FZB42 could modify plant root development, hence improving the plant nutrients’ uptake. A considerable increase was noticed in the root length, root tips, root average diameter, root average volume, and surface area treated with pure *Bacillus* VOCs, as compared to the CK. These findings predicted that the *Bacillus* strain (GBAC46, RJGP41, and FZB42)-produced VOCs promoted plant growth primarily by influencing tomato root morphogenesis. 

Plants have evolved an enzymatic defense mechanism, in which antioxidant enzymes play a key role [59]. When plants recognize any pathogen attack or biotic and abiotic stress, these antioxidant enzymes are activated, and a strong response is exerted against the pathogen attack or stress [60]. Previously, it was well-documented that VOCs produced by *Bacillus* spp. reduced the pathogen infection on lychee fruit and regulated the enzymatic activity by minimizing the adverse impact of oxidative stress produced after pathogen infection [20,61]. Previous studies reported the induction of antioxidant enzymes and protein contents after pathogen invasion [2,18,19], but very little is known about the role of VOCs on PGP. In our study, we found that when plants are exposed to VOCs (2P3E, CBDO, and BDH), antioxidant enzymes (CAT, SOD, POD, and APX) are significantly activated, resulting in a reduction of oxidative stress after biotic or abiotic stress. Furthermore, the results showed that the application of pure VOCs significantly enhanced the protein content in tomato plants compared to the CK. It is common knowledge that ROS tends to accumulate in environments with high-salt and heavy-metal stresses. As a result, oxidative damage caused by increased oxidative stress decreased the protein content [62].

Microbial VOCs are important regulators in several signaling processes and pathways of gene regulation against different plant diseases [27]. Previous studies show a substantial increase in the relative expression levels of *SICKX1, EXP18*, and *SIAA1* [63]. Furthermore, when *Arabidopsis thaliana* was exposed to VOCs, an increase in the relative expression of the expansion genes (*EXP1, EXPB3, EXPB5,* and *EXP5*) was observed [56]. In our experiment findings, the selected *Bacillus* pure VOCs, 2P3E, CBDO, and BDH, were also observed to upregulate growth-promoting genes such as *IAA4* (associated with auxin-response protein production)*, ARF10A* (associated with auxin-response factor 10A)*, CKX2* (associated with cytokinin oxidase/dehydrogenase)*, GA2OX2* (associated with gibberellin 2-oxidase), and *EXP1* (associated with expansion protein production). Our experiment results showed that the expression of the *ERF* gene involved in the ethylene pathway was downregulated upon *Bacillus* pure VOC exposure, according to the previous findings [18,19,64].

## 5. Conclusions

The study concludes that the *Bacillus* isolates (GBAC46, RJGP41, and the positive control FZB42 VOCs) had a significant role in the PGP of tomato seedlings. The *Bacillus* VOCs, CBDO, 2P3E, and BDH, also had a crucial role in plant growth-promoting factors. The pure VOCs increased the plant growth-promoting parameters in both in vitro and in-pot experiments. Furthermore, our study provided new insights on the mechanism of *Bacillus* VOCs involved in the regulation of antioxidant enzyme activity and total protein contents, as well as the significant genes involved in PGP in tomato plants. 

## Figures and Tables

**Figure 1 biology-12-00940-f001:**
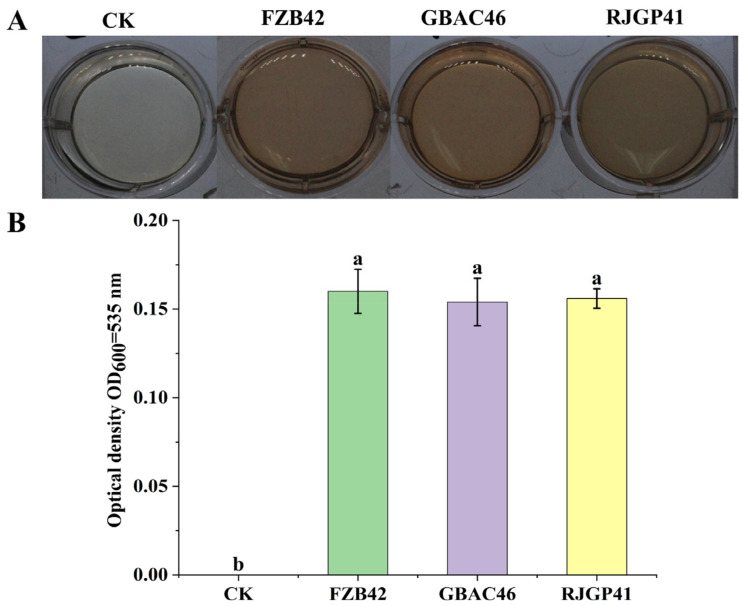
*Bacillus* isolates showing IAA production in culture broth at OD_600_ = 535 nm. (**A**) Visual IAA production of selected isolates in a 12 well-microplate. (**B**) Graphical representation of IAA production of each *Bacillus* isolate. The error bars represent each treatment’s mean and standard deviation. The lowercase letters above the error bars show a significant difference. Tukey’s HSD test was used to calculate significant differences between treatments at *p* ≤ 0.05.

**Figure 2 biology-12-00940-f002:**
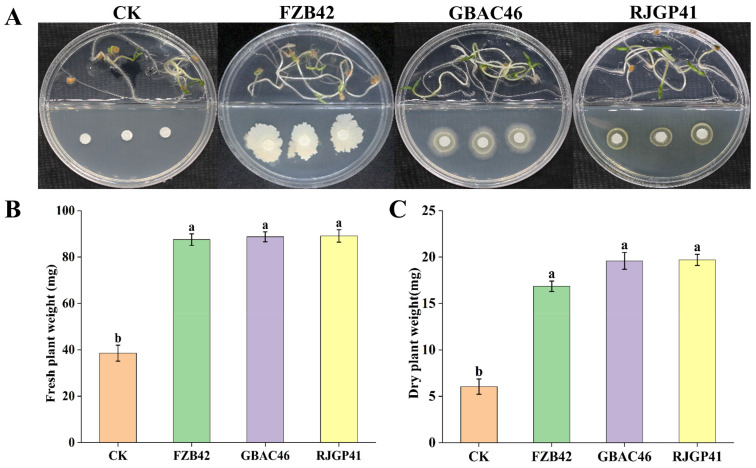
The effect of *Bacillus* VOCs on tomato seedlings. (**A**) Visual representation of seedling growth promotion. Physiological parameters: (**B**) fresh seedlings’ weight (g) and (**C**) dry seedlings’ weight (g). The error bars represent each treatment’s mean and standard deviation. The lowercase letters above the error bars show a significant difference. Tukey’s HSD test was used to calculate the significant differences between treatments at *p* ≤ 0.05.

**Figure 3 biology-12-00940-f003:**
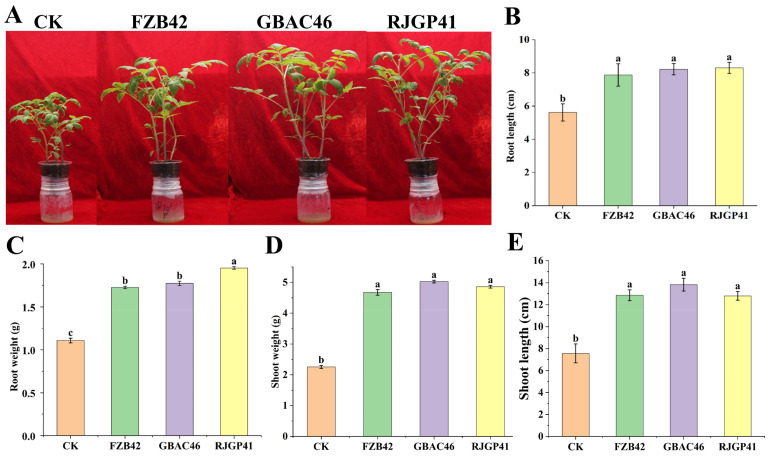
Effect of *Bacillus* VOCs on PGP attributes in tomato. (**A**) Pictorial presentation of the effect of *Bacillus* VOCs on tomato plant growth. Graphical presentation of the *Bacillus* VOCs on physiological traits: (**B**) root length (cm), (**C**) root weight (g), (**D**) shoot weight (g), and (**E**) shoot length (cm). The error bars represent each treatment’s mean and standard deviation. The lowercase letters above the error bars show a significant difference. Tukey’s HSD test was used to analyze the significant differences between treatments at *p* ≤ 0.05.

**Figure 4 biology-12-00940-f004:**
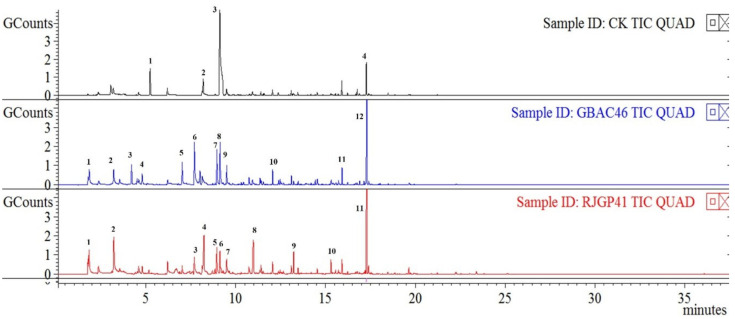
The chromatogram of the possible GBAC46 and RJGP41 VOCs detected through GC-MS analysis. VOCs detected in both strains and the control were excluded. The VOCs were selected at time intervals ranging from 0 to 30 min. The numbering shows peaks for different VOCs.

**Figure 5 biology-12-00940-f005:**
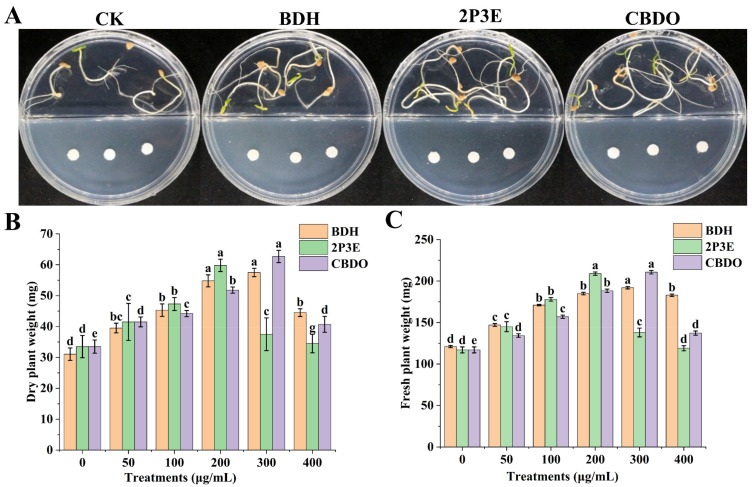
Role of pure VOCs on PGP of tomato plants. (**A**) Visual PGP of tested pure VOCs. (**B**,**C**) Graphical presentation of the pure VOCs BDH, 2P3E, and CBDO. Tukey’s HSD test calculated the significant differences between different treatments at *p* ≤ 0.05. The lowercase letters above the error bars show a significant difference. The error bars represent the mean and standard deviation of three replicates, with three repeats for each treatment.

**Figure 6 biology-12-00940-f006:**
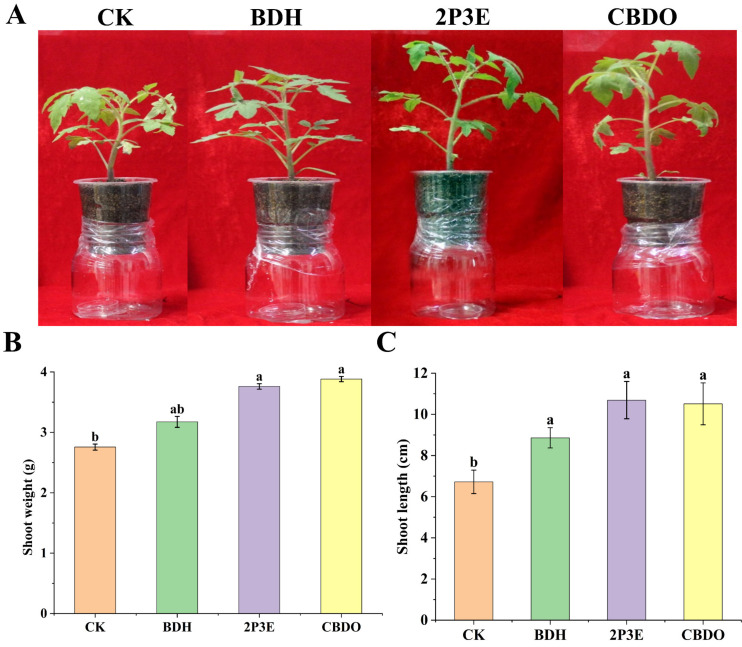
Effect of pure VOCs on tomato PGP. (**A**) Pictorial presentation of pure VOCs on PGP in tomato. (**B**) Graphical presentation of the pure VOCs on root length (cm) and (**C**) shoot weight (g). The error bars represent the mean and standard deviation for each treatment with three replicates. The lowercase letters above the error bars show a significant difference. Tukey’s HSD test was used to calculate the significant differences between treatments at *p* ≤ 0.05.

**Figure 7 biology-12-00940-f007:**
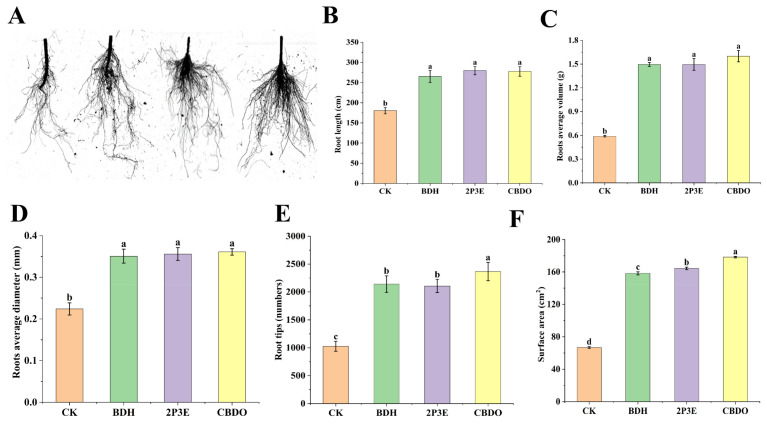
The rhizo-scanning image of tomato roots exposed to BDH, 2P3E, and CBDO-VOCs in a greenhouse experiment. (**A**) The visual expression of pure VOCs on root morphology. (**B**) The effect of pure VOCs on root length (cm), (**C**) root average volume (g), (**D**) root average diameter (mm), (**E**) root tips (numbers), and (**F**) surface area (cm^2^). The error bars show each treatment’s mean (n = 5) and standard deviation. The lowercase letters above the error bars show a significant difference. Tukey’s HSD test was used to calculate the significant differences between treatments at *p* ≤ 0.05 and the experiment was carried out in triplicate.

**Figure 8 biology-12-00940-f008:**
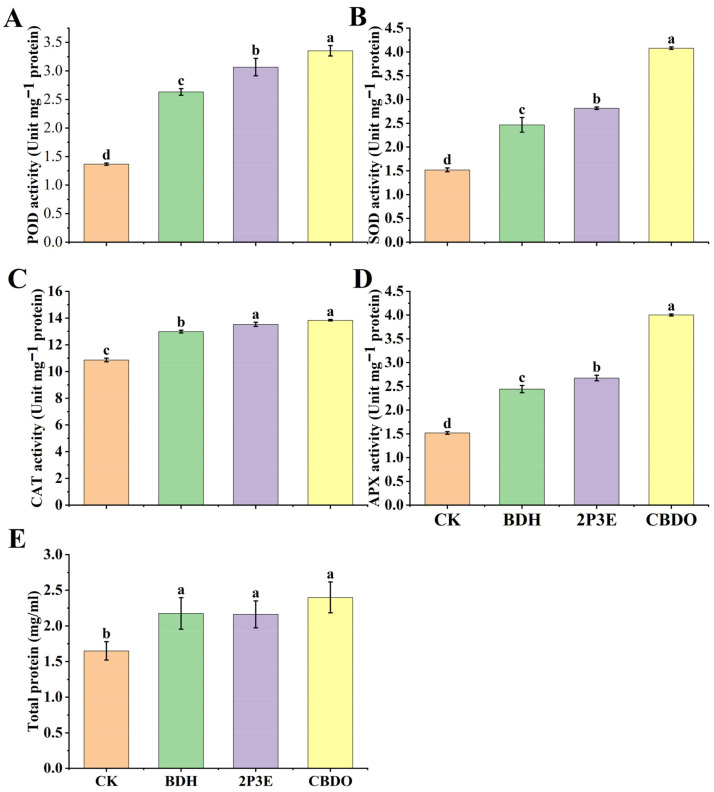
The effect of the pure VOCs 2P3E, CBDO, and BDH on antioxidant enzymes in tomato leaves. (**A**) Peroxidase (POD), (**B**) superoxide dismutase (SOD), (**C**) catalase (CAT), (**D**) ascorbate peroxidase (APX), and (**E**) total protein contents. The leaf samples of different treatments were taken after 5 days of exposure to pure VOCs to obtain the enzyme extract. The error bars show each treatment’s mean (n = 3) and standard deviation. The lowercase letters above the error bars show a significant difference. Tukey’s HSD test was used to calculate the significant differences between treatments at *p* ≤ 0.05 and the experiment was performed in triplicate.

**Figure 9 biology-12-00940-f009:**
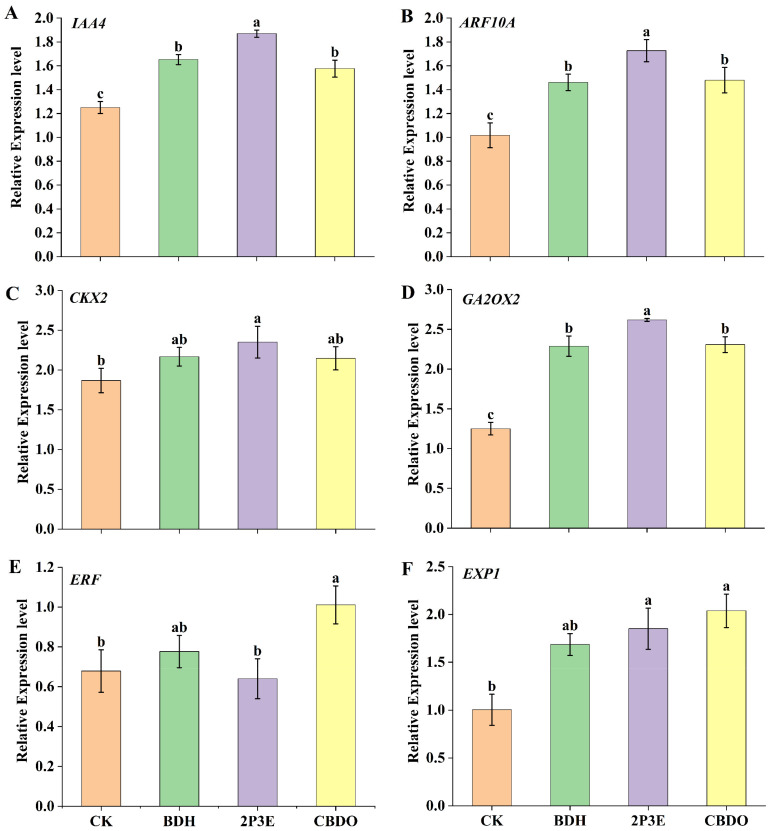
The expression of growth-promoting genes in tomato plants following 5 days of exposure to pure VOCs (BDH, 2P3E, and CBDO). (**A**) *IAA4*, (**B**) *ARF10A*, (**C**) *CKX2*, (**D**) *GA2OX2*, (**E**) *ERF*, and (**F**) *EXP1*. Tukey’s HSD test was used to distinguish mean values at *p* ≤ 0.05. The lowercase letters above the error bars show a significant difference. The error bars represent the mean and standard deviation for each treatment.

**Table 1 biology-12-00940-t001:** VOCs detected through GC-MS analysis of *Bacillus* strains GBAC46 and RJGP41.

Sr.#	GBAC-46 VOCs Detected via GCMS	Probability	MW	CAS#	Formula
1	Undecanal, 2-methyl	23.51	184	110-41-8	C_12_H_24_O
2	1-4-pentadiene	18.01	68	591-93-5	C_5_H_8_
3	L-threonine	28.59	119	72-19-5	C_4_H_9_NO_3_
4	Amiodarone	8.54	645	1951-25-3	C_25_H_29_I_2_NO_3_
5	Dl-2-Aminoadipic acid	47.06	161	542-32-5	C_6_H_11_NO_4_
6	2-pentanone, 3-methyl	76.30	100	565-61-7	C_6_H_12_O
7	3-Hexanamine, 3-ethyl	40.72	129	56667-17-5	C_8_H_19_N
8	1-dodecanamine, N, N dimethyl	13.95	213	112-18-5	C_14_H_31_N
9	2-Heptanone, 6-methyl	48.24	128	928-68-7	C_6_H_16_O
10	2-Undecanone	37.56	170	112-12-9	C_11_H_22_O
11	Pentadecane	16.19	212	629-62-9	C_15_H_32_
12	2-Tridecanone	36.07	198	593-08-08	C_13_H_26_O
**RJGP41**					
1	Cyclobutane 1,2,3,4-tetramethyl	31.15	112	69531-57-3	C_8_H_16_
2	1,3-Cyclobutanediol,2,2,4,4-tetramethyl	63.31	144	3010-96-6	C_8_H_16_O_2_
3	3 Hexanamine, 3-ethyl	11.68	129	56667-17-5	C_8_H_19_N
4	2 Hexanone, 5-methyl	26.78	114	110-12-3	C_7_H_14_O
5	2 Heptanone	23.03	114	110-43-0	C_7_H_14_O
6	2 Heptanone, 6-methyl	46.25	128	928-68-7	C_8_H_16_O
7	Undecanal, 2-methyl-thyl	30.73	184	110-41-8	C_12_H_24_O
8	1,1,3,3,5,5,7,7,9,9, decamethyl-9 (2 methylpropoxy) pentasiloxane-1-01	15.10	444	NONE	C_14_H_40_O_6_Si_5_
9	Choleston-3-one cyclic 1,2-ethane diyl-actal (5-beta)	22.03	430	25328-53-4	C_29_H_50_O_2_
10	Tridecane	12.14	184	629-50-5	C_13_H_28_
11	Octasiloxane,1,1,3,3,5,5,7,7,9,9,11,11,13,13,15,15-hexadecamethyl	42.41	5781	19095-24-0	C_16_H_50_O_7_Si_8_

## Data Availability

The data supporting reported results can be found in the main text and in Appendix A.

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
