# Peer review of "Plant–Microbes Interaction: Exploring the Impact of Cold-Tolerant Bacillus Strains RJGP41 and GBAC46 Volatiles on Tomato Growth Promotion through Different Mechanisms"

_biology, 2023, doi:10.3390/biology12070940_

Round 1
Reviewer 1 Report
-
Full form of acronyms is required whenever it being introduced for the first time in the manuscript. (for eg: PGPR)
-
Determination of antioxidant enzyme activity
-
Control was water instead of non exposed leaf. It is not an appropriate control for this experiment. Extract from non-exposed leaf should be used to set the baseline activity of thee enzymes.
- Protein quantification data is needed to ensure equal amount of enzyme was present in extracts tested for activity.
-
Please describe how the enzyme activity was tested ? The following line (207) - The absorbance activity of each enzyme was recorded at 470 nm for POD, 560 nm for SOD, 240 nm for CAT, and 290 nm for APX - does not give enough clarity of the assay ? This information points towards absorbance and the graph in figure 8 mentions activity. What is the baseline activity in plants not exposed to VOC ?
-
Figure 3 description - Graph labelling (i.e. B, C, D, E) do not match with text below the figure.
-
What does ‘a’, ‘b’ etc on the error bars stand for ? Please clearly mention in the methods section
Moderate improvements to sentence structure and grammar is needed for this manuscript.
Author Response
Point to Point response to reviewers 1:
Response; Thank you very much for your helpful suggestions and valuable input in our research manuscript. We also very much appreciate the comments/suggestions made by referees. According to the suggestions, we have modified/changed and did major/minor improvements during revision throughout the manuscript. We also incorporated most of the suggestions during our revision. Accordingly, a point-by-point response is provided below. And the revisions are highlighted in the main text with red and blue color and track changes.
Full form of acronyms is required whenever it being introduced for the first time in the manuscript. (for eg: PGPR)
Response; Thanks, the abbreviation has been modified.
Determination of antioxidant enzyme activity
Control was water instead of non-exposed leaf. It is not an appropriate control for this experiment. Extract from non-exposed leaf should be used to set the baseline activity of thee enzymes.
Response; Thanks, the sentence has been. Actually, we used leaf exposed to ddH2O was used as a control, ddH2O was not used as a control itself.
Protein quantification data is needed to ensure equal amount of enzyme was present in extracts tested for activity.
Response; Thanks, the protein quantification experiment will be done in a future study.
Please describe how the enzyme activity was tested ? The following line (207) - The absorbance activity of each enzyme was recorded at 470 nm for POD, 560 nm for SOD, 240 nm for CAT, and 290 nm for APX - does not give enough clarity of the assay ? This information points towards absorbance and the graph in figure 8 mentions activity. What is the baseline activity in plants not exposed to VOC ?
Response; Thanks, the antioxidant material and methods has been described in details.
Figure 3 description - Graph labelling (i.e. B, C, D, E) do not match with text below the figure.
Response; Thanks, the legend of figure has been changed.
What does ‘a’, ‘b’ etc on the error bars stand for ? Please clearly mention in the methods section
Response; Thanks, the small letters has been explained in the M&M sections.

Reviewer 2 Report
Dear Editor
The research article entitled "Plant-Microbes Interaction: Exploring the Impact of Cold Tolerant Bacillus Strains RJGP41 and GBAC46 Volatiles on Tomato Growth Promotion Through Different Mechanisms" by Khan et., al have briefly described by cold-tolerant Bacillus strains RJGP41 and GBAC46 from the Qinghai-Tibet Plateau and the well-known PGPR strain FZB42 and their VOCs on tomato plants. Moreover, authors revealed that both Bacillus isolates and their pure VOCs positively improve PGPR activities in tomato plants by triggering antioxidant enzyme activity and expression of the PGPR genes. The overall manuscript is well-written, there are several typing and spelling mistakes, however, the concerns about this manuscript can be found below and minor revision is suggested.
Kindly revise the following corrections
Abbreviations should be written as full terms (abbreviation) when used for the first time in the text.
Line (29), replace VOCs recognized with VOCs identified.
Line (65-69) Rewrite the sentence, firstly direct impact should be mentioned and then indirect impact issues should be written.
Reference citations are not according to mdpi format and authors need to follow the mdpi author guidelines.
Line (50-52) authors write ((High Killing Rate of Nematode and Promotion of Rice Growth by Synthetic Volatiles from Bacillus Strains Due to Enhanced Oxidative Stress Response, n.d.) What does it mean? I think authors cited references in wrong format.
Line (98-99) same wrong reference citations.
Line (109-110) same wrong reference citations.
Line (103-104), the sentence, “Furthermore, the study also adds new information about the mechanism of Bacillus VOCs by regulating the key genes involved in promoting tomato plant growth” is confusing at the end, rewrite the sentence with easy understanding.
Line (117) Put a full stop at the end of the sentence.
Line (131) the author mentioned Partitioned (compartment) Petri dishes but in the Abstract Section Line (32) the author mentioned partitioned (I-plate). Use either a compartment petri plate or an I-plate in the manuscript. Don’t use both names.
Line (139-141) the author mentioned “After incubation, tomato seedlings were evaluated for growth promotion activities (fresh plant weight and dry plant weight)”. Growth promotion includes fresh and dry plant weight so no need to mention activities.
Line (175) Rewrite the word chemicals identified or VOCs identified with chemicals/VOCs detected in the whole manuscript text.
Line (177-178) replace was with were as VOCs are plural and remove the word activity.
In lines (306-308) the author mentioned he used VOCs concentrations (50, 100, 200, 300, and 400μg/mL) but in Fig (5) author also used 0 concentration. Add the concentration to the main results.
Line (343-344) the author mentioned a considerable increase in root morphological studies, the word studies is a little confusing. Rephrase the sentence for proper understanding.
Line (384-394) in section ‘3.8 Effect of pure VOC on plant growth promotion genes’ the author did not mention how much fold gene expression was upregulated or downregulated. It should be mentioned in folds.
Line 426 the word (substancial is wrong is should be a substantial).
Authors need to recheck the scientific name and gene names and it should be in italic format.
Author Response
Point to Point response to reviewers 2:
The research article entitled "Plant-Microbes Interaction: Exploring the Impact of Cold Tolerant Bacillus Strains RJGP41 and GBAC46 Volatiles on Tomato Growth Promotion Through Different Mechanisms" by Khan et., al have briefly described by cold-tolerant Bacillus strains RJGP41 and GBAC46 from the Qinghai-Tibet Plateau and the well-known PGPR strain FZB42 and their VOCs on tomato plants. Moreover, authors revealed that both Bacillus isolates and their pure VOCs positively improve PGPR activities in tomato plants by triggering antioxidant enzyme activity and expression of the PGPR genes. The overall manuscript is well-written, there are several typing and spelling mistakes, however, the concerns about this manuscript can be found below and minor revision is suggested.
Response; Thank you very much for your helpful suggestions and valuable input in our research manuscript. We also very much appreciate the comments/suggestions made by referees. According to the suggestions, we have modified/changed and did major/minor improvements during revision throughout the manuscript. We also incorporated most of the suggestions during our revision. Accordingly, a point-by-point response is provided below. And the revisions are highlighted in the main text with red and blue color and track changes.
Kindly revise the following corrections
Abbreviations should be written as full terms (abbreviation) when used for the first time in the text.
Response; Thanks, the abbreviation has been modified.
Line (29), replace VOCs recognized with VOCs identified.
Response; Thanks, the word has been changed.
Line (65-69) Rewrite the sentence, firstly direct impact should be mentioned and then indirect impact issues should be written.
Response; Thanks, the sentence has been modified.
Reference citations are not according to mdpi format and authors need to follow the mdpi author guidelines.
Response; Thanks, the reference has been modified.
Line (50-52) authors write ((High Killing Rate of Nematode and Promotion of Rice Growth by Synthetic Volatiles from Bacillus Strains Due to Enhanced Oxidative Stress Response, n.d.) What does it mean? I think authors cited references in wrong format.
Response; Thanks, the reference has been modified.
Line (98-99) same wrong reference citations.
Line (109-110) same wrong reference citations.
Response; Thanks, the reference has been changed and modified according to journal guidelines.
Line (103-104), the sentence, “Furthermore, the study also adds new information about the mechanism of Bacillus VOCs by regulating the key genes involved in promoting tomato plant growth” is confusing at the end, rewrite the sentence with easy understanding.
Response; Thanks, we have modified the sentence.
Line (117) Put a full stop at the end of the sentence.
Response; Thanks, we have modified.
Line (131) the author mentioned Partitioned (compartment) Petri dishes but in the Abstract Section Line (32) the author mentioned partitioned (I-plate). Use either a compartment petri plate or an I-plate in the manuscript. Don’t use both names.
Response; Thanks, the sentence has been modified.
Line (139-141) the author mentioned “After incubation, tomato seedlings were evaluated for growth promotion activities (fresh plant weight and dry plant weight)”. Growth promotion includes fresh and dry plant weight so no need to mention activities.
Response; Thanks, the has been modified.
Line (175) Rewrite the word chemicals identified or VOCs identified with chemicals/VOCs detected in the whole manuscript text.
Response; Thanks, the sentence has been changed.
Line (177-178) replace was with were as VOCs are plural and remove the word activity.
Response; Thanks, the word has been replaced.
In lines (306-308) the author mentioned he used VOCs concentrations (50, 100, 200, 300, and 400μg/mL) but in Fig (5) author also used 0 concentration. Add the concentration to the main results.
Response; Thanks, the result has been explained.
Line (343-344) the author mentioned a considerable increase in root morphological studies, the word studies is a little confusing. Rephrase the sentence for proper understanding.
Response; Thanks, the sentence has been rephrased.
Line (384-394) in section ‘3.8 Effect of pure VOC on plant growth promotion genes’ the author did not mention how much fold gene expression was upregulated or downregulated. It should be mentioned in folds.
Response; Thanks, the sentence has been modified and explained.
Line 426 the word (substancial is wrong is should be a substantial).
Response; Thanks, the words has been changed
Authors need to recheck the scientific name and gene names and it should be in italic format.
Response; Thanks, the all scientific name and genes has been changed in italic.

Reviewer 3 Report
The authors have to address the following points
1- The references should be reformatted according to MDPI style
2- There should be a positive control for the growth promotion effects like IAA in a suitable dose for example to compare the described effects
3- Scale bars are required for most of the photos
4- Language of the manuscript requires further editing
Moderate editing of English language required
Author Response
Point to Point response to reviewers 3:
Response; Thank you very much for your helpful suggestions and valuable input in our research manuscript. We also very much appreciate the comments/suggestions made by referees. According to the suggestions, we have modified/changed and did major/minor improvements during revision throughout the manuscript. We also incorporated most of the suggestions during our revision. Accordingly, a point-by-point response is provided below. And the revisions are highlighted in the main text with red and blue color and track changes.
The references should be reformatted according to the MDPI style
Response; Thanks, the reference has been modified.
There should be a positive control for the growth promotion effects like IAA in a suitable dose for example, to compare the described effects. For IAA estimation, one positive and one negative control was used. FZB42 was used as a positive control whereas culture broth was used as a negative control.
Response; Thanks, we have used FZB42 as a positive control for all experiments.
Scale bars are required for most of the photos
Response; Thanks, the photos has been taken with camera so are not able to add scale bars.
The language of the manuscript requires further editing
Response; Thanks, the language has been improved during the revision of MS.

Round 2
Reviewer 1 Report
1) For the following comment I made in my review -
"What does ‘a’, ‘b’ etc on the error bars stand for ? Please clearly mention in the methods section" - the authors gave following response :
"Response; Thanks, the small letters has been explained in the M&M sections"
However, I do not see any mention of this in the methods section (2.10) of the new manuscript sent to me (attached below). If you have indeed made this correction, convey the line numbers to me.
2) Without protein quantification data, the antioxidant enzyme activity data does not hold any value. Without knowing how much of the specific protein is being used in the assay, one can't speculate on the magnitude of it's activity. For instance, does the enzyme activity increase because there is simply more enzyme in your extracted sample or is it because the enzyme kinetics has changed as a result of your treatment. If you wish to show protein quantification date in another study then the entire section about antioxidant enzyme activity (section 3.7 and Figure 8)has to be removed from this manuscript.

English is fine. Could be further improved.
Author Response
Point to Point response to reviewers 1:
For the following comment I made in my review -
"What does ‘a’, ‘b’ etc on the error bars stand for? Please clearly mention in the methods section" - the authors gave following response:
"Response; Thanks, the small letters have been explained in the M&M sections"
However, I do not see any mention of this in the methods section (2.10) of the new manuscript sent to me (attached below). If you have indeed made this correction, convey the line numbers to me.
Response; Thanks, the small letters (a b and c) have been explained in each figure of legends.
2) Without protein quantification data, the antioxidant enzyme activity data does not hold any value. Without knowing how much of the specific protein is being used in the assay, one can't speculate on the magnitude of it's activity. For instance, does the enzyme activity increase because there is simply more enzyme in your extracted sample or is it because the enzyme kinetics has changed as a result of your treatment. If you wish to show protein quantification date in another study, then the entire section about antioxidant enzyme activity (section 3.7 and Figure 8) has to be removed from this manuscript.
Response; Thanks, well raised questions. We have done the protein content activity in our experiment and it has been explained according to reviewer suggestions.

Reviewer 3 Report
I am pleased to accept the manuscript in its present form
Author Response
I am pleased to accept the manuscript in its present form
Response: Thank you very much for your valuable time to review our manuscript.
Round 3
Reviewer 1 Report
Changes made are satisfactory.